# Differential impacts of reduced worktime on work-life balance in Korea

**Hyun Ju Kim[1], Hye Myung Lee[2], Heejoo Cheon[3], Hansoo Ko[4]***

**1** Institute on Disability, University of New Hampshire, Durham, New Hampshire, United States of America, **2** Department of Sociomedical Sciences, Columbia University Mailman School of Public Health, New York, New York, United States of America, **3** Schar School of Policy and Government, George Mason University, Fairfax, Virginia, United States of America, **4** Department of Health Administration and Policy, George Mason University, Fairfax, Virginia, United States of America

* hko20@gmu.edu

**Data Availability Statement:** The data files are available and can be fully downloaded at the Korea Labor Institute website (https://www.kli.re.kr/klips_eng).

**Funding:** The authors received no specific funding for this work.

## Abstract

This study analyzes the heterogenous effects of reducing weekly work hour on workers' quality of life in Korea. Using longitudinal household data from the Korean Labor and Income Panel Study (KLIPS) from 2001 to 2017, this study aims to shed light on how the work hour reduction policy may differently affect workers with different levels of resources and support by demographic and socioeconomic status. Our estimates from the difference-in-differences approach exploiting the staggered implementation of the work hour reduction policy indicate that the policy increased leisure satisfaction of female workers with low educational level and female workers in regular or inflexible work setting. Given Korea's exceedingly long working hours and inequities in the labor market, a better understanding of the complex factors that affect work-life balance can be helpful in developing policies supportive of healthy work-life balance for workers. This study, to our knowledge, is the first to investigate the composition of workers in the Korean labor market and examine differential impacts of the workhour reduction policy by demographic and socioeconomic status.

## Introduction

Long working hours have negative impacts on workers' health, safety, and work-life balance [1–9]. A recent systematic review reports that workhour reduction interventions improve workers' well-being [10]. Among the OECD countries, South Korea had the fourth longest working hours in 2008, which were 210 hours more than the OECD average annual hours. In 2022, South Korea still works 149 hours more than the OECD average [11]. To address the excessively long working hours, the Korean government introduced the Five-Day Working System which cut down the weekly work limit from 44 to 40 hours. Between 2004 and 2011, the system was implemented in stages based on the industry and the establishment size of a company, aiming to improve the quality of life, increase worker creativity, and enhance national competitiveness [1, 12–14].

The 40-hour workweek policy was first introduced in the public, financial, and insurance sectors and workplaces with 1,000 or more employees. Then, the workweek standard of 40 hours was further extended to the remaining industries in stages by the size of the

**Competing interests:** The authors have declared that no competing interests exist.

establishments (with 300+, 100+, 50+, and 20+ workers). As of 2011, the new workweek limit was effective in all establishments with at least 5 or more employees. Note that, as workers can work maximum 12 hours of overtime work and 16 hours of Saturday/Sunday work, there were still workers working up to 68 hours a week even under the new regulation [15].

While working hours have been an important research area for labor studies, the impacts of reduced hours of work on workers' quality of life have not been comprehensively studied [10]. Previous research on Korea's weekly workhour reduction found that the reduced workhour increased productivity and safety of the workplace [14, 16], as well as enhanced individual's health status [1, 13] and decreased family conflict [17]. Similarly, studies in other countries suggest that reduced working hours can improve an individual's quality of life by enhancing their mental well-being [18], reducing work-life conflict [9], relieving health symptoms [19, 20], and lowering health and safety risks [1, 21]. However, there is limited evidence on the mechanism of how long working hours negatively affect workers' quality of life [10]. As the impact of workhour reduction on quality of life depends not only on the absolute amount of time devoted to leisure activities but on the individual's social context, the reduction of worktime is expected to have differential impacts on the quality of life depending on the factors affecting an individual's time constraints such as gender, socioeconomic status, and work environment.

Gender is among the main factors predicting job-related inequity and determining work-life balance of workers [22, 23]. Some studies on the gendered impacts of long working hours have mainly focused on marital status and family commitments [24–26]. In particular, Rudolf [25] explored the impact of working hour reduction on the subjective well-being among Korean parents with children by gender but did not find significant results. The impact of the workhour reduction policy on quality of life can also differ by an individual's socioeconomic status [27–30]. However, empirical predictions on the impacts of the reduced working hours depending on socioeconomic status remain ambiguous. Socioeconomically advantaged individuals may be more likely to benefit from the reduced work hours as they may have better access to leisure facilities and activities [31]. On the other hand, individuals with limited resources and support may benefit from a marginal reduction in their working time, as they may have a lower workhour threshold at which their well-being deteriorates [32]. Thus, this study distinguishes the effects of work reduction by gender and explores the interplays between gender and socioeconomic status.

Working conditions also relate to the effects of reduced working hours and are becoming emerging policy challenges in Korea. Since the 1997 financial crisis, the share of non-regular (mainly part-time and temporary) workers significantly and consistently rose in the nation [33]. A 2011 government survey indicated that Korean workers with precarious employment tend to be women and less-educated [34]. In addition, average wages of female workers with irregular working arrangements were only about 50 percent of male regular workers' wages in 2018 [35]. Thus, given the increasing labor market flexibility, research on the link between work arrangements and workers' quality of life is crucial in Korea. We expect that the impacts of workhour reduction are different by working environment as well as gender and socioeconomic status. First, precarious employment, which refers to work arrangements that are uncertain, unstable, or insecure [36, 37], can have negative impacts on work-life balance. Workers in precarious employment may have less control over their work schedules and may be required to work irregular or unpredictable hours, which can make it difficult to plan and manage their personal and family commitments. In addition, workers in precarious employment may have less access to employee benefits, such as paid time off and sick leave, which can further disrupt their work-life balance [38, 39].

Predictions on the impacts of flexible work time, which refers to work arrangements that allow employees to vary their start and end times within certain limits [40], are also unclear. On the one hand, flexible work time can provide workers with greater control over their work schedules and allow them to better align their work and personal commitments [41–43]. On the other hand, flexible working hours can also create additional demands and expectations, such as the need to be available outside of traditional working hours or to work longer hours in order to make up for the time spent outside of the office. This can lead to increased workloads and blurring boundaries between work and personal life, which can harm work-life balance [44]. Consistent with this prediction, a descriptive study of the 1998 French law reducing workhours reported that the policy's positive effect on employees' work-life balance was substantially bigger among those with standard working hours than peers with non-standard working time [27]. Additionally, several studies report the flexible worktime results in negative perception on female workers [45–47].

We aim to address the gap in literature by shedding light on how the work hour reduction policy may differently affect workers with different levels of resources and support. South Korea offers an interesting case study due to its consistently long working hours and policy efforts to reduce working hours. Exploiting a quasi-experimental setting provided by the staggered implementation of the Five-Day Working System in Korea, this study is, to our knowledge, the first to examine heterogenous effects of workhour reduction on workers' quality of life. This study is distinguished from previous research by using leisure satisfaction as a measure of quality of life, rather than job satisfaction and life satisfaction [25]. Leisure satisfaction captures an individual's level of enjoyment and fulfillment in their leisure activities, which can have a significant impact on their overall well-being and happiness. Previous studies have included selected leisure attributes, such as the amount of non-work time, spare time activities, and access to leisure facilities, in their assessments on quality of life [29, 48–50]. For example, Osborne [51] found that life quality includes both the "conditions of life" and the "experience of life," and leisure attributes can play a role in both aspects of life quality.

The remainder of this study is organized as follows: Section II discusses the data sources and methodology being used. Section III presents the results, which are followed by a discussion in Section IV. Section V concludes the study and discusses the policy implications.

## Method

### Data

This study uses longitudinal data from the Korean Labor and Income Panel Study (KLIPS) from 2001 to 2017. The KLIPS has surveyed nationally representative samples of urban households and their members on labor force participation and labor market activities since 1998. A wide range of information at the individual level, including employment status, job characteristics, working hours, demographic and socioeconomic characteristics, and life/leisure satisfaction, is collected each year. In particular, the KLIPS contains complete information on establishment sizes of each surveyed worker's workplaces, which enabled us to construct a policy indicator on whether the workhour reduction policy was in effect at workplaces where individuals were working.

We restrict our study sample to the working age population (aged 20 to 64) who earns non-zero wage income in each wave. To construct a sample of those who are most likely to be affected by the 40-hour workweek policy, we restrict our sample to full-time wage earners by excluding those working 30 hours or fewer per week. Note that, as the KLIPS does not provide information on contracted work hours, we cannot differentiate work hours and contracted hours in our analysis. We also exclude individuals who experience a reverse in their policy

status (e.g., changing jobs to a smaller establishment thus exempted from the 40-work hour limit) from our analysis. We also maximize the chance of fully capturing the impact of Five-Day Working System by excluding survey years 2018 or later, as additional reduction in the legal maximum working time was implemented in 2018 from 68 hours to 52 hours. As a result, our main study sample (N = 54,449 person-years) consists of an unbalanced panel of 5,619 men and 4,659 women.

## Empirical strategy

Exploiting the stepwise implementation of reduced worktime policy by establishment size (Table 1), we construct a policy indicator ($Policy_{it}$) that takes 1 if the policy was in effect at workplaces where the individual was working. Our control group consists of workers never affected by the policy and those not affected by the policy yet. We specify the following difference-in-differences framework with two-way fixed effects linear regressions:

$$Y_{it} = \alpha_{it} + \beta Policy_{it} + \gamma_i + \delta_t + X_{it} + \epsilon_{it} \tag{1}$$

$\beta$ is our parameter of interest which is the difference-in-differences coefficient on the impacts of the 40-hour workweek policy. In our two-way fixed effects model, we add individual fixed effects ($\gamma_i$) to control for time-invariant and unobservable factors at the individual level. Year fixed effects ($\delta_t$) are included to address unobserved time-related socioeconomic changes which are constant across all individuals. $X_{it}$ represents covariates including 10-year age group dummies (ages 20–29, 30–39, 40–49, 50–59, and 60–65) and a marriage indicator for individual i at time t. To address region-specific economic changes that may be associated with the working environment, Gross Regional Domestic Product (GRDP) per capita of the region is included based on where the individual resides.

Following Bryson and Blanchflower [52], we also control for survey month fixed effects. To adjust differences in access to leisure activities and employment opportunities across groups, fixed effects on the region of residence (16 state dummies) and the standard classification of occupational groups by Korean Standard Classification of Occupations system are included in our main specification. The occupational groups include 'managers', 'professionals and related workers', 'clerks', 'service workers', 'sales workers', 'skilled agricultural, forestry, and fishery workers', 'craft and related trades workers', 'equipment, machine operating and assembling workers', and 'elementary workers'.

To check the robustness of our results, we run the analysis using different samples. First, we relax our sample restriction by including wage earners working more than 20 hours per week. Also, a sample of survey years spanning 2001 through 2012 was used to address potential bias from differential attrition where follow-up loss may happen more in the control group. The results of the robustness checks appear in S1 Table. In addition, we restricted our sample to employees who reported staying within the same establishment size during 2004–2011 in

**Table 1. Staggered implementation of 40-hour standard weekly work policy.**

| Effective date | Establishment size and industry |
| --- | --- |
| July 1, 2004 | Financial sector, public sector, establishments with 1000+ employees |
| July 1, 2005 | Establishments with 300 to 999 employees |
| July 1, 2006 | Establishments with 100 to 299 employees |
| July 1, 2007 | Establishments with 50 to 99 employees |
| July 1, 2008 | Establishments with 20 to 49 employees |
| July 1, 2011 | Establishments with 5 to 19 employees |

order to address potential selection bias. The results are reported in S2 Table. and discussed at the end of the result section.

## Variables

Our outcome variable ($Y_{it}$) consists of both worktime measures and satisfaction measures. Work time indicates the individual's regular weekly working hours. Given the possibility of skewness of the continuous work time measure, we also test the policy impacts on an indicator whether the individual worked more than 40 hours a week. Satisfaction measures include three binary satisfaction indicators (workhour satisfaction, job satisfaction, and leisure satisfaction). Each satisfaction measure is based on single items, respectively asking "How satisfied are you with your working hours in relation to your main job?", "How satisfied are you overall with your main job?", and "How satisfied are you with your leisure activities?" Satisfaction is coded as 1 if the respondent answered 'extremely satisfied' or 'satisfied' to each satisfaction questionnaire from a 5-point Likert scale in the survey. We dichotomized the satisfaction measures because the extreme measures have smaller cell sizes (all less than 1 percent in our analytic sample). Therefore, combining the responses simplifies computational processes. The questionnaires of the work hours and satisfaction measures are translated to English and presented in S3 Table.

This study conducts multiple subgroup analysis based on the respondent's demographic and socioeconomic status including gender, educational attainment, marital status, parental status, employment precarity, and flexible employment. Gender is defined as male or female. Educational attainment is categorized as 'high school completion or less' or 'college level education or above' using the respondents' highest level of education achieved. Marital status is identified as 'married' or 'unmarried'. Parental status is specified as having at least one child or not. Following Kim et al. (2008), we assign precarious employment if a worker works daily, temporarily, or part-time. We define flexible employment as not having fixed regular working hours on a contract or arranging working hours via informal negotiation by employee and line manager, rather than through organization's formal process [53, 54].

## Results

Table 2 reports summary statistics of the sample of full-time workers at the baseline period in 2011. Male workers on average work longer hours (53.8 hours) than female workers (47.8 hours) per week with a higher likelihood of working over 40 hours a week. Perhaps expectedly, workhour satisfaction of male workers is lower (21.6 percent) than female (27.3 percent), while job satisfaction appears similar in both groups. Higher proportion of male workers (16 percent) report leisure satisfaction than female workers (14.5 percent). Female workers in the baseline period are younger, while male workers are more likely to be married and have children. A greater proportion of male workers have college level education or above (38.9 percent) than female workers (34.9 percent). It is notable that more female workers are in precarious (36.3 percent) and flexible employment settings (26.2 percent).

As shown in Fig 1, our data confirms that the work hour reduction policy decreased the average weekly work hours and ratio of overtime workers since the inception of the policy in 2004. The reduction effects are observed both in male and female workers, while female workers overall work fewer hours with a less likelihood of working over 40 hours a week than the counterpart.

Table 3 presents general effects of reduced work hours on all workers (Panel A), full-time male workers (Panel B), and full-time female workers (Panel C). Column (1) and (2) show the policy's direct effect on work hours, while columns (3) through (5) present the effects on

**Table 2. Descriptive statistics of the sample at baseline period (2001).**

| | (1) | (2) | (3) |
|---|---|---|---|
| | All | Male | Female |
| Weekly working hours | 51.4 | 53.8 | 47.8 |
| Working over 40 hours a week (%) | 84.8 | 89.0 | 78.4 |
| Workhour satisfaction (%) | 23.9 | 21.6 | 27.3 |
| Job satisfaction (%) | 20.6 | 20.2 | 21.2 |
| Leisure satisfaction (%) | 15.4 | 16.0 | 14.5 |
| Ages 20–29 (%) | 27.2 | 19.6 | 38.6 |
| Ages 30–39 (%) | 30.5 | 35.8 | 22.5 |
| Ages 40–49 (%) | 26.3 | 26.6 | 25.8 |
| Ages 50–59 (%) | 12.7 | 14.4 | 10.0 |
| Ages 60–64 (%) | 3.4 | 3.5 | 3.2 |
| Married (%) | 67.6 | 74.7 | 56.7 |
| Have children (%) | 51.8 | 55.3 | 46.5 |
| College level education or above (%) | 37.3 | 38.9 | 34.9 |
| Precarious employment (%) | 27.1 | 21.1 | 36.3 |
| Flexible employment[a] (%) | 23.0 | 21.0 | 26.2 |
| GRDP per capita (in 1,000 won) | 15,096 | 15,134 | 15,037 |
| N | 3,672 | 2,216 | 1,456 |

[a] Summary statistics reflect averages in 2005, since the information on flexible employment was available from 2005.

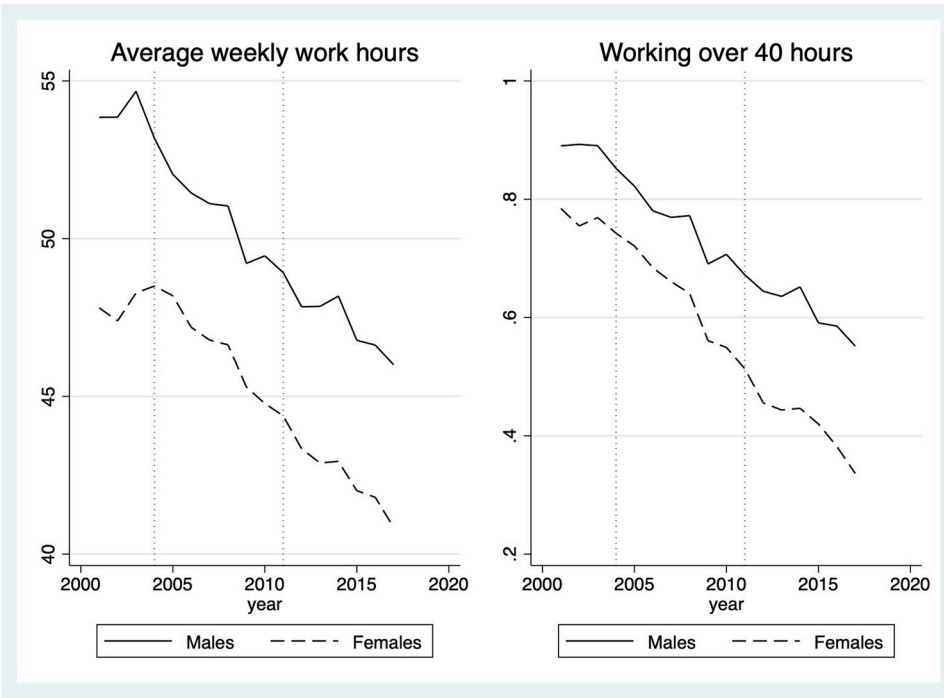

**Fig 1. Trends in average weekly working hours and the percentage of working over 40 hours per week, by sex.**
Note: Dotted vertical lines indicate years of 2004 and 2011 which represent the start and end years of the staggered policy implementation, respectively. The graph on the left-hand side shows the yearly trends in average weekly work hours (continuous variable) by sex. The graph on the right-hand side shows the yearly trends of ratio of workers (%) who work over 40 hours a week by sex.

**Table 3. Effects of reduced work hours.**

| | (1) | (2) | (3) | (4) | (5) |
|---|---|---|---|---|---|
| | **Weekly workhours** | **Workhours>40 weekly** | **Job Satisfaction** | **Workhour Satisfaction** | **Leisure Satisfaction** |
| Panel A: All Workers (N = 54,449) | | | | | |
| Policy implementation | -0.970*** (0.176) | -0.060*** (0.007) | 0.042*** (0.008) | 0.036*** (0.008) | 0.011 (0.008) |
| Control group mean | 53.444 | 0.866 | 0.235 | 0.280 | 0.199 |
| Panel B: Male Workers (N = 34,101) | | | | | |
| Policy implementation | -1.051*** (0.231) | -0.053*** (0.009) | 0.034*** (0.010) | 0.031*** (0.010) | 0.003 (0.010) |
| Control group mean | 55.092 | 0.903 | 0.223 | 0.255 | 0.199 |
| Panel C: Female Workers (N = 20,348) | | | | | |
| Policy implementation | -0.764*** (0.268) | -0.068*** (0.012) | 0.056*** (0.013) | 0.046*** (0.014) | 0.024** (0.012) |
| Control group mean | 51.116 | 0.814 | 0.251 | 0.316 | 0.199 |

*** $p<0.01$,

** $p<0.05$,

* $p<0.1$

Each cell represents separate regression. Regressions also control for age groups, marital status, gross regional domestic production per capita, survey year and month dummies, job type dummies, provincial dummies, and individual fixed effects. Robust standard errors in parentheses.

satisfaction. Note that each column represents the estimates of the two-way fixed effects linear regressions specified in Eq (1). On average, male workers (55.09 hours per week) have longer work hours than female workers (51.12 hours per week). In addition, among the control group, the share of workers with 40 or longer weekly work hours was higher among male workers (90.3 percent) than female workers (81.4 percent). As shown in column (1), the work hour reduction policy decreased weekly work hours by 0.97 hours in total workers, while male workers experienced a greater reduction in work hour (1.05 hours) than female workers (0.76 hours). The 40-hour workweek policy led to a 6-percentage point reduction in the probability of a full-time worker to work more than 40 hours a week (Column (2)). This reduction in the likelihood of overtime work was significant and meaningful for both male and female full-time workers. As mentioned in the introduction, because workers may work up to 68 hours (including 12 hours of overtime work and 16 hours of Saturday/Sunday work) per week even after the policy change [15], the workhour reduction policy did not reduce the share of overtime work to zero.

The reduction in workhour led to significant increases in job and workhour satisfaction (Columns (3) and (4)). After the policy was implemented, the probability that a full-time worker was satisfied with his/her job went up by 4.2 percentage points, and the likelihood of workhour satisfaction increased by 3.6 percentage points. These effects were significant for both male and female full-time workers. In Column (5), the estimated impact of the policy on the probability of being satisfied with leisure time was positive (1.1 percentage points increase), though the coefficient was insufficiently different from zero. However, we find heterogenous effects on leisure satisfaction where the work hour reduction led to a significant increase in leisure satisfaction only among female full-time workers by 2.4 percentage points. This effect was notable in that the implementation of the 40-hour workweek policy increased female workers' leisure satisfaction by 12.1percent, compared to the control group mean.

To further explore heterogeneity in the effects of the workhour reduction policy, we first show a subgroup analysis by gender and educational attainment in Table 4. Columns (1) and (2) confirm that workers' weekly workhours are generally reduced across gender and educational attainment. Results in Columns (3) and (4) indicate that all subgroups show significant

**Table 4. Effects of reduced work hours: Subgroup analysis by gender and educational attainment.**

| | (1) | (2) | (3) | (4) | (5) |
| --- | --- | --- | --- | --- | --- |
| | Weekly workhours | Workhours>40 weekly | Job Satisfaction | Workhour Satisfaction | Leisure Satisfaction |
| Panel A: Male Workers with High School Completion or Less (N = 15,542) | | | | | |
| Policy implementation | -1.151*** (0.346) | -0.035*** (0.011) | 0.039*** (0.013) | ** (0.013) | -0.007 (0.013) |
| Control group mean | 56.938 | 0.916 | 0.147 | 0.186 | 0.147 |
| Panel B: Male Workers with College or Higher (N = 18,559) | | | | | |
| Policy implementation | -0.730** (0.308) | -0.065*** (0.013) | 0.025* (0.015) | 0.033** (0.015) | 0.011 (0.015) |
| Control group mean | 52.667 | 0.885 | 0.323 | 0.345 | 0.267 |
| Panel C: Female Workers with High School Completion or Less (N = 11,292) | | | | | |
| Policy implementation | -0.333 (0.384) | -0.055*** (0.015) | 0.035** (0.015) | 0.042** (0.017) | 0.041*** (0.014) |
| Control group mean | 52.945 | 0.844 | 0.186 | 0.249 | 0.135 |
| Panel D: Female Workers with College or Higher (N = 9,053) | | | | | |
| Policy implementation | -0.990*** (0.354) | -0.069*** (0.021) | 0.082*** (0.022) | 0.044* (0.023) | -0.001 (0.022) |
| Control group mean | 47.778 | 0.757 | 0.371 | 0.440 | 0.315 |

*** p<0.01,

** p<0.05,

* p<0.1

Each cell represents separate regression. Regressions also control for age groups, marital status, gross regional domestic production per capita, survey year and month dummies, job type dummies, provincial dummies, and individual fixed effects. Robust standard errors in parentheses.

impacts on job and workhour satisfaction. Notably, no workers in different subgroups but female workers with low educational attainment (high school diploma or less) experienced enhanced leisure satisfaction (Column (5)). Specifically, the workhour reduction policy enhanced the level of leisure satisfaction for female workers with lower educational attainment by 4.1 percentage points (30.4 percent increase compared to control group mean). Given the gap in leisure satisfaction among the female control groups (13.5 percent in the lower education group as opposed to 31.5 percent in the higher education group), this result suggests that the implementation of the 40-hour workweek policy narrowed inequity in life quality among female workers.

We report differential impacts by marital status in Table 5. Columns (1) and (2) show that all subgroups saw reductions in overtime work although the impact was smaller for unmarried male workers. With respect to job and workhour satisfaction, male and female workers display different tendencies by marital status in Columns (3)-(4). Specifically, married male workers appear to have substantial gains in job (3.4 percentage points) and workhour satisfaction (3.9 percentage points) than unmarried peers, whereas among the female workers, the impacts were significant for unmarried female workers (8.1 percentage points in job satisfaction and 6.9 percentage points in workhour satisfaction) than married female workers. The impact on leisure satisfaction appears to be positive for married male workers and both married and unmarried female workers in Column (5), although the estimates are insignificant.

As parental status can be an additional factor in gendered impact of work hours on quality of life [24, 25], we present a subgroup analysis by gender and parental status in Table 6. Columns (1) and (2) confirm that all subgroups across gender and parental status worked fewer

**Table 5. Effects of reduced work hours: Subgroup analysis by gender and marital status.**

| | (1) | (2) | (3) | (4) | (5) |
|---|---|---|---|---|---|
| | Weekly workhours | Workhours>40 weekly | Job Satisfaction | Workhour Satisfaction | Leisure Satisfaction |
| Panel A: Unmarried Male Workers (N = 9,417) | | | | | |
| Policy implementation | -0.135 (0.467) | -0.033* (0.017) | 0.018 (0.019) | 0.014 (0.020) | -0.004 (0.018) |
| Control group mean | 54.736 | 0.887 | 0.188 | 0.234 | 0.166 |
| Panel B: Married Male Workers (N = 24,684) | | | | | |
| Policy implementation | -1.511*** (0.279) | -0.059*** (0.010) | 0.034*** (0.012) | 0.039*** (0.012) | 0.003 (0.012) |
| Control group mean | 55.244 | 0.909 | 0.238 | 0.264 | 0.213 |
| Panel C: Unmarried Female Workers (N = 8,130) | | | | | |
| Policy implementation | -0.426 (0.435) | -0.057*** (0.019) | 0.081*** (0.022) | 0.069*** (0.023) | 0.031 (0.021) |
| Control group mean | 51.124 | 0.832 | 0.235 | 0.313 | 0.211 |
| Panel D: Married Female Workers (N = 12,218) | | | | | |
| Policy implementation | -0.869** (0.362) | -0.062*** (0.016) | 0.035** (0.016) | 0.028 (0.018) | 0.022 (0.016) |
| Control group mean | 51.111 | 0.801 | 0.263 | 0.318 | 0.190 |

*** $p < 0.01$,

** $p < 0.05$,

* $p < 0.1$

Each cell represents separate regression. Regressions also control for age groups, gross regional domestic production per capita, survey year and month dummies, job type dummies, provincial dummies, and individual fixed effects. Robust standard errors in parentheses.

hours after the introduction of the workhour reduction policy (0.74 hours to 1.3 hours). Similar to the marital status results (Table 4), however, male and female workers show different patterns in job and work hour satisfaction depending on their parental status. Among male full-time workers, those with children experienced greater increases in job and workhour satisfaction (both by 4.6 percentage points) than male workers without children (2.7 percentage points in job satisfaction). In contrast, the impacts on job and work hour satisfaction were more distinct among female workers without children (5.7 and 5.2 percentage points) than peers with children (4.6 percentage points in job satisfaction). We find no significant estimates in leisure satisfaction, although the size of the impact appears to be larger in female workers than male counterparts. However, as shown in Tables 5 and 6, we did not find evidence of heterogenous effects of the reduction in working hours on leisure satisfaction by marital status and parental status.

We now show how different employment characteristics affect the set of outcome variables depending on the gender of workers. First, as precarious employment provides less autonomy over working hours to employees, Table 7 presents heterogenous effects by precarious employment on male and female workers. Consistent with our expectation, both male and female workers without precarious employment experienced a significant reduction in their weekly work hours (1.08 hour decrease for male workers and 0.98-hour increase for female workers, respectively). Also, the probability of overtime work decreased only for workers without precarious employment by 6 to 8 percentage points (Column (2)). The policy change also increased job and workhour satisfaction among male workers without precarious employment (3.7 and 2.7 percentage points) and job satisfaction of female workers (5.4 percentage points). The effect on workhour satisfaction of female workers was also positive with a similar effect size to the male counterpart, though statistically insignificant.

As another indicator of employment characteristics, the flexibility of worktime is used to show the heterogenous impacts of the reduced work week policy in Table 8. Similar to the

**Table 6. Effects of reduced work hours: Subgroup analysis by gender and parental status.**

| | (1) | (2) | (3) | (4) | (5) |
|---|---|---|---|---|---|
| | Weekly workhours | Workhours>40 weekly | Job Satisfaction | Workhour Satisfaction | Leisure Satisfaction |
| Panel A: Male Workers without Children (N = 16,546) | | | | | |
| Policy implementation | -0.765** (0.368) | -0.050*** (0.013) | 0.027* (0.015) | 0.019 (0.016) | 0.003 (0.015) |
| Control group mean | 55.207 | 0.892 | 0.211 | 0.249 | 0.198 |
| Panel B: Male Workers with Children (N = 17,555) | | | | | |
| Policy implementation | -1.310*** (0.330) | -0.045*** (0.013) | 0.046*** (0.014) | 0.046*** (0.015) | 0.001 (0.014) |
| Control group mean | 54.982 | 0.913 | 0.235 | 0.260 | 0.199 |
| Panel C: Female Workers without Children (N = 11,991) | | | | | |
| Policy implementation | -0.772** (0.373) | -0.074*** (0.016) | 0.057*** (0.018) | 0.052*** (0.019) | 0.020 (0.017) |
| Control group mean | 51.621 | 0.823 | 0.236 | 0.305 | 0.213 |
| Panel D: Female Workers with Children (N = 8,357) | | | | | |
| Policy implementation | -0.735* (0.418) | -0.060*** (0.020) | 0.046** (0.021) | 0.023 (0.022) | 0.011 (0.019) |
| Control group mean | 50.437 | 0.801 | 0.272 | 0.331 | 0.179 |

*** $p<0.01$,

** $p<0.05$,

* $p<0.1$

Each cell represents separate regression. Regressions also control for age groups, marital status, gross regional domestic production per capita, survey year and month dummies, job type dummies, provincial dummies, and individual fixed effects. Robust standard errors in parentheses.

previous results, both male and female workers without flexible work time experienced decreases in their weekly work hours (Column (1) of Panels B and D). Similar patterns are observed in the impacts on the likelihood of overtime work (Columns (2)). Job and workhour satisfaction are enhanced in male workers without flexibility of worktime by 2.8 percentage points and 2.6 percentage points, respectively. Among the subgroups, only female workers without flexible worktime saw an increase in leisure satisfaction by 3.8 percent (15.7 percent higher than the control group mean), indicating that they were among primary beneficiaries of the reduced work hour policy.

Results of the robustness check analyses are discussed. First, we restrict our main sample to wage earners who work more than 30 hours per week to assess the policy impact on full-time workers. If workers who work less than 30 hours per week had responded differently to the policy, our results may have been sensitive to working hours. However, when we expanded our sample to include wage earners who work more than 20 hours per week (S1 Table Column (1)), the results were similar to our main findings. Second, our study period spans 17 years which leaves possibilities of follow-up loss. In Column (2) of S1 Table, we present our results using a shorter study period (2001–2012) and the estimates are similar to our main results.

Furthermore, our results may be driven by workers who could self-select into establishments of particular sizes that benefit from the reduced work hours. In order to address the self-selection bias, we restricted our sample to workers who reported the same establishment size (within three categories: smaller than 50, 50–500, and 500 employees or more) during 2004–2011. This robustness check also found similar results S2 Table while standard errors are increased slightly due to the smaller sample size.

**Table 7. Effects of reduced work hours: Subgroup analysis by gender and employment characteristics: Precarious employment.**

| | (1) | (2) | (3) | (4) | (5) |
|---|---|---|---|---|---|
| | Weekly workhours | Workhours>40 weekly | Job Satisfaction | Workhour Satisfaction | Leisure Satisfaction |
| Panel A: Male Workers with Precarious Employment (N = 7,001) | | | | | |
| Policy implementation | -0.531 (0.619) | -0.038 (0.024) | 0.014 (0.020) | 0.056** (0.023) | 0.037* (0.020) |
| Control group mean | 54.157 | 0.819 | 0.115 | 0.155 | 0.104 |
| Panel B: Male Workers without Precarious Employment (N = 27,100) | | | | | |
| Policy implementation | -1.084*** (0.255) | -0.058*** (0.010) | 0.037*** (0.012) | 0.027** (0.012) | -0.002 (0.012) |
| Control group mean | 55.351 | 0.926 | 0.253 | 0.283 | 0.225 |
| Panel C: Female Workers with Precarious Employment (N = 6,924) | | | | | |
| Policy implementation | 0.174 (0.637) | -0.037 (0.024) | 0.028 (0.023) | 0.036 (0.025) | -0.003 (0.019) |
| Control group mean | 52.000 | 0.756 | 0.155 | 0.226 | 0.130 |
| Panel D: Female Workers without Precarious Employment (N = 13,424) | | | | | |
| Policy implementation | -0.977*** (0.294) | -0.076*** (0.016) | 0.054*** (0.018) | 0.029 (0.018) | 0.040** (0.017) |
| Control group mean | 50.661 | 0.843 | 0.301 | 0.363 | 0.224 |

*** $p<0.01$,

** $p<0.05$,

* $p<0.1$

Each cell represents separate regression. Regressions also control for age groups, gross regional domestic production per capita, survey year and month dummies, job type dummies, provincial dummies, and individual fixed effects. Robust standard errors in parentheses.

## Discussion

Previous research found that the reduction in weekly workhour in Korea contributed to enhanced labor productivity [14], lower industrial injury [1], less marital dissolution [17], healthier behavior among workers [16], and reduced obesity [13]. However, this study is the first to highlight the composition of workers in the Korean labor market and examine differential impacts of the workhour reduction policy by demographic and socioeconomic status. Using a quasi-natural experiment setting in the change of Korea's weekly workhour policy and longitudinal data, we find that the workhour-reduction policy had heterogenous effects on quality of life, particularly by gender and education. Specifically, low-educated female workers were more likely than college-educated peers to benefit from workhour reduction, narrowing the gap in leisure satisfaction between the two groups to some extent; the estimated impact on quality of life among low-educated female workers is about 23 percent of the difference in control group means between the two groups. Put differently, for college-educated female workers, a significant reduction in working time did not translate to an increased leisure satisfaction, while female workers with less education experienced a greater leisure satisfaction after the introduction of the work reduction policy. This finding is consistent with the proposition by Dinh et al. [32] that individuals with limited resources would benefit more from workhour reduction because they have a lower working hour threshold at which their quality of life deteriorates. This result also supports the findings of Korpi et al. [55] that presented an increased labor participation by female workers without college education from a host of family policies in the OECD countries. Highly educated female workers are more likely to have a

**Table 8. Effects of reduced work hours: Subgroup analysis by gender and employment characteristics: Flexible worktime.**

| | (1) | (2) | (3) | (4) | (5) |
|---|---|---|---|---|---|
| | **Weekly workhours** | **Workhours>40 weekly** | **Job Satisfaction** | **Workhour Satisfaction** | **Leisure Satisfaction** |
| Panel A: Male Workers with Flexible Worktime (N = 6,228) | | | | | |
| Policy implementation | -0.747 (0.667) | -0.039 (0.025) | -0.012 (0.023) | 0.054** (0.024) | 0.009 (0.024) |
| Control group mean | 55.964 | 0.842 | 0.161 | 0.156 | 0.142 |
| Panel B: Male Workers without Flexible Worktime (N = 20,906) | | | | | |
| Policy implementation | -0.836*** (0.290) | -0.040*** (0.012) | 0.028** (0.014) | 0.026* (0.015) | -0.007 (0.014) |
| Control group mean | 53.621 | 0.868 | 0.259 | 0.303 | 0.229 |
| Panel C: Female Workers with Flexible Worktime (N = 3,701) | | | | | |
| Policy implementation | 0.162 (0.999) | 0.048 (0.035) | 0.049 (0.035) | 0.090** (0.036) | -0.002 (0.029) |
| Control group mean | 52.370 | 0.748 | 0.180 | 0.215 | 0.140 |
| Panel D: Female Workers without Flexible Worktime (N = 12,288) | | | | | |
| Policy implementation | -1.010*** (0.318) | -0.057*** (0.017) | 0.049*** (0.018) | 0.027 (0.019) | 0.038** (0.017) |
| Control group mean | 49.844 | 0.757 | 0.308 | 0.380 | 0.242 |

*** $p<0.01$,

** $p<0.05$,

* $p<0.1$

Each cell represents a separate regression. Regressions also control for age groups, gross regional domestic production per capita, survey year and month dummies, job type dummies, provincial dummies, and individual fixed effects. Robust standard errors in parentheses.

choice to work or not without the help of family friendly policies because they tend to have more resources and support at the individual level.

On the work environment factors, we find that workers in precarious employment or with flexible time arrangements did not experience a significant decline in working hours after the policy was implemented. This result indicates that the policy resulted in an unintended consequence favoring workers with standard work arrangement. Given the increased labor market flexibility and a growing body of evidence on negative impacts of long working hours [1–9], this differential impact would have widened labor market inequities in Korea. With respect to work environment factors and leisure satisfaction, it is notable that only female workers without precarious or flexible employment had higher leisure satisfaction after the introduction of the workhour reduction policy. This result supports Kim's [54] finding that female workers' life satisfaction is increased when flexible worktime is formally arranged through the organizational process. It is because workers are more likely to feel forced to work more if the flexible work arrangement is informally negotiated with managers. This indicates that the work reduction policy may be effective only for workers with regular working conditions, thus limiting benefits to workers in marginalized work environments.

Finally, our findings suggest that the decline in working hours led to significant increases in job satisfaction and workhour satisfaction for most subgroups. However, a significant workhour reduction did not necessarily translate to increased leisure satisfaction but affected the quality of life depending on educational attainment. These findings suggest that the impact of working hours on workers' well-being meaningfully varies by socioeconomic status. In fact, previous studies found that the relationship between working hours and life satisfaction can

vary by labor market structures [56], gender and parental status [57, 58], and cross-partner dynamics and culture [25, 59]. Perhaps future research on individuals' time use may better inform the mechanism through which working hours affect an individual worker's time constraints.

Taken together, these findings are particularly relevant to Korea's recent discussion on the labor policies. We confirmed that the work hour reduction policy generated heterogenous effects on workers by gender, educational levels, and work environment. Our findings suggest that a modest decline in working time can be an effective policy option addressing inequities in the labor market. At the same time, however, the work hour reduction can only benefit workers with regular work schedule arrangement. Therefore, it is unclear which group of workers the work reduction policy might have benefitted. Nevertheless, the Korean government's recent policies to address gender inequities in the labor market (such as providing maternity leave and childcare centers, incentivizing employers to hire back women) lack in taking heterogeneity among female workers into consideration [60]. Thus, these findings call for changes in the government's approach toward policies properly targeting vulnerable populations such as female workers with low education level or workers in precarious or flexible work environment, with a better view of heterogeneity in workers' socioeconomic status. Given the Korean government's recent announcement to abolish the limit on the maximum weekly working hours [61], this study underscores the role of labor policy tailored for heterogenous workers and their quality of life. A better understanding of the complex factors that affect work-life balance can be helpful in developing policies supportive of healthy work-life balance for workers.

Limitations of this study are acknowledged. First, this study is based on self-reported weekly working hours of workers. While self-reported working hours can be susceptible to recall bias, we were not able to use official working hours for each worker due to data availability. Second, this study uses satisfaction outcome measures which can be subjective by each individual. Due to data availability, we were not able to incorporate objective measures which may complement the shortcomings of solely using subjective measures. Future studies using other relevant outcome measures (e.g., time-use measures) are needed. Third, this study is based on three satisfaction measures, including job, workhour, and leisure satisfaction. As these three measures may not fully represent the satisfaction of other aspects in one's life, we ran the analysis using three additional satisfaction measures in S4 Table (family relationship satisfaction, housing satisfaction, and social life satisfaction). We find that the satisfaction measure on family relationship increased significantly among women with low educational attainment, which aligns with our main findings. We also find that housing satisfaction significantly increased among female workers without precarious employment or without flexible employment. Fourth, we acknowledge that our analysis includes multiple subgroups, and there is an increased possibility of finding significant estimates by chance due to small sample sizes within the subgroups. While this possibility would warrant further research with more complete data, the patterns of significant results are consistent with our theoretical framework and support the findings from prior research [22–24, 26–30]. Fifth, as our data do not contain workplace identifiers, we are not able to detect changes in workhours of the control group that may have been affected by unobservable workplace-specific factors. Given the nature of enforcing working hour reduction, it is natural to imagine that the workhour-reduction effects could have varied by workplaces (i.e. heterogeneity by share of highly-educated workers). Thus, we chose to present results from two-way fixed effects regressions as our main specification and show additional regression results using an alternative difference-in-differences estimation suggested by de Chaisemartin and D'Haultfoeuillle [62], which requires a carefully selected control group, in S5 Table. Overall, results are qualitatively similar to the results from our main specification.

## Conclusion

Worktime is one of the essential factors in enhancing worker's quality of life [57, 63]. This study analyzed effects of Korea's weekly workhour reduction policy on a set of satisfaction outcomes and found differential impacts by demographic and socioeconomic status. This indicates that no one-size-fits-all labor policies may achieve policy goals, particularly in a labor market with heterogenous workers characteristics. Given the inequities in the Korean labor market and non-worker-friendly policy environment, our findings suggest developing a workhour policy that focuses on helping marginalized workers' quality of life.

Future studies can investigate why reduced working hours impacted certain women's leisure satisfaction groups. Since this study only utilized quantitative methods, conducting a mixed-methods analysis may provide a better understanding of the mechanism of the reduced working hours policy on those with different socioeconomic statuses and specific demographic characteristics. Additionally, future research can explore working hours policies across countries with other labor market structures. Comparative studies can provide insights on the generalizability of the findings and the potential for policy transfer.

## Supporting information

**S1 Table. Robustness checks.**
(DOCX)

**S2 Table. Self-selection check.**
(DOCX)

**S3 Table. Variable description.**
(DOCX)

**S4 Table. Other satisfaction measures.**
(DOCX)

**S5 Table. Generalized difference-in-differences estimates.**
(DOCX)

## Author Contributions

**Conceptualization:** Hyun Ju Kim, Hye Myung Lee, Heejoo Cheon, Hansoo Ko.

**Data curation:** Hyun Ju Kim, Hye Myung Lee, Hansoo Ko.

**Formal analysis:** Hyun Ju Kim, Hye Myung Lee, Heejoo Cheon, Hansoo Ko.

**Investigation:** Hyun Ju Kim, Hye Myung Lee, Heejoo Cheon, Hansoo Ko.

**Methodology:** Hyun Ju Kim, Hye Myung Lee, Heejoo Cheon, Hansoo Ko.

**Project administration:** Hyun Ju Kim, Heejoo Cheon, Hansoo Ko.

**Resources:** Hansoo Ko.

**Supervision:** Hansoo Ko.

**Validation:** Hyun Ju Kim, Hye Myung Lee, Heejoo Cheon, Hansoo Ko.

**Visualization:** Hyun Ju Kim, Hye Myung Lee.

**Writing – original draft:** Hyun Ju Kim, Hye Myung Lee, Heejoo Cheon, Hansoo Ko.

**Writing – review & editing:** Hyun Ju Kim, Hye Myung Lee, Heejoo Cheon, Hansoo Ko.

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
