## [Decision Letter · Decision Letter 0]

25 Apr 2023

PONE-D-23-04903Differential Impacts of Reduced Worktime on Work-Life Balance in KoreaPLOS ONE

Dear Dr. Ko,

Thank you for submitting your manuscript to PLOS ONE. After careful consideration, we feel that it has merit but does not fully meet PLOS ONE’s publication criteria as it currently stands. Therefore, we invite you to submit a revised version of the manuscript that addresses the points raised during the review process.

We look forward to receiving your revised manuscript.

Kind regards,

Tae-Young Pak, Ph.D.

Academic Editor

PLOS ONE

Reviewers' comments:

Reviewer's Responses to Questions

**Comments to the Author**

1. Is the manuscript technically sound, and do the data support the conclusions?

Reviewer #1: Partly

Reviewer #2: Partly

2. Has the statistical analysis been performed appropriately and rigorously? 

Reviewer #1: No

Reviewer #2: Yes

3. Have the authors made all data underlying the findings in their manuscript fully available?

Reviewer #1: Yes

Reviewer #2: No

4. Is the manuscript presented in an intelligible fashion and written in standard English?

Reviewer #1: Yes

Reviewer #2: Yes

5. Review Comments to the Author

Reviewer #1: The paper is generally well written and concise. It addresses an important topic of work hour regulations, work hours and well being among employees in different demographic groups.

Method

An impressive dataset spanning 16 years, with the regulations implemented at different time points, allowing the authors to distinguish between development over time and implementation.

The authors exclude those working 30 hours or fewer per week. Is it possible to distinguish on contracted work hours rather than actual work hours?

The author’s state: “Individual fixed effects (γi) control for time-invariant and unobservable factors at the individual level”. Does this imply a) that the authors have used a fixed effects regression, analyzing only within individual differences? If yes, specify. If not, clarify.

The questions used for the key variables are missing. I particularly miss the wording of the questions for work hours and Satisfaction (translated to English), and if these questions are self-made or developed and tested elsewhere. Furthermore, is satisfaction measured using single items or scales? if scales pleas also add alpha values.

I also miss information on the types of analyses, and how to interpret the coefficients in table 3. Can I assume that model 1 (weekly work hours) is a linear regression, while the rest is logistic regression? And are the coefficients for logistic regression presented OR, average marginal effects or what?

Satisfaction is measured on a 5 point Likert scale, but the authors dichotomize the outcome. Why? Keeping the variation in the outcome is generally preferable, and alternative solutions should be explained.

Results and discussion

Several comparisons are made between different subgroups. The authors have done multiple subgroups analyses – stratifying by gender, education, marital status etc. However, the authors have not tested for interaction. I would recommend testing for interactions when making comparisons regarding effects in different groups. If refraining from doing so the authors should be careful in their interpretations as they have not tested if the effect is indeed significantly different in their subgroups.

For example: when the authors state “whereas the impacts were stronger for unmarried female workers (8.1 percentage points in job satisfaction and 6.9 percentage points in workhour satisfaction)» they have not actually tested if the impact is significantly stronger for unmarried female workers. It is fully possible that the effect size is significant in one group, and not in another without the two groups having significantly different effect sizes. It is also fully possible that two groups both have significant effects sizes, but that the effect sizes are still significantly different.

Similarly, the authors state in the discussion “These findings suggest that socioeconomic status is among the main factors moderating the impact of working hours on workers’ well-being.” But they have not run a moderation analyses.

Notably, if the authors use fixed affects analyses (only analyzing within effects) interaction between unchanging characteristics is not straight forward. But alternative methods can also take advantage of the longitudinal dataset and allow for interaction effects.

The authors should also acknowledge the consequences of testing each relationship in 26 (4 x 6 + 2) different subgroups. The share number of analyzes increases the possibility that the size and significance of the coefficients will vary by chance.

Languish

Some minor languish errors e.g. “… confirm that workers’ weekly workhours generally reduced across” - sentence might be missing an “are”

Reviewer #2: The aim of this paper is to analyze the impact of reducing weekly work hours on workers' life satisfaction in Korea, using longitudinal household data. Specifically, the study aims to investigate how the work hour reduction policy may affect workers with different demographic and socioeconomic backgrounds. The findings from the difference-in-differences approach indicate that the policy increased leisure satisfaction among female workers with low educational levels.

While the topic of this paper is important and interesting, there are some limitations that need to be addressed, including the need for a broader set of measures for life satisfaction, addressing the endogeneity problem, and providing a more thorough discussion of the identification strategy. Furthermore, the interpretations of the findings should be enhanced to draw robust policy implications from the results of this study.

1. The authors only consider 'job satisfaction', 'work hours satisfaction', and 'leisure satisfaction' as measures of quality of life. It is not clear why other aspects of life satisfaction, such as 'satisfaction with family members or with social relationships', and 'satisfaction with living environment', as well as ‘overall life satisfaction’ are not included in the analysis. Additionally, the differential impacts across different aspects of life satisfaction need to be interpreted more thoroughly.

2. The endogeneity problem is a major concern in this study. The policy was implemented gradually from 2004 to 2011, and workers may have self-selected into establishments of certain sizes to benefit from the reduction of work hours. This could violate the exogeneity assumption of the model, as transitions between establishments before the policy was implemented may not be random. This issue needs to be addressed in the analysis.

3. I would like to see a more thorough discussion of the identification strategy. What variations are taken as exogenous using the staggered implementation of the new policy? When the biggest size group (1,000 or more workers) underwent the policy change, should we think of the rest of the size groups as the control, synonymous to a diff-in-diff strategy? I encourage the author to refer to the papers such as Goodman-Bacon (2021), and explain how we should interpret the results given that the firm-size groups received treatment at different times

6. PLOS authors have the option to publish the peer review history of their article (what does this mean?). If published, this will include your full peer review and any attached files.

Reviewer #1: No

Reviewer #2: No

---

## [Author Response · Author response to Decision Letter 0]

18 Sep 2023

Please see the rebuttal letter file for our responses to reviewers.

---

## [Decision Letter · Decision Letter 1]

30 Oct 2023

Differential Impacts of Reduced Worktime on Work-Life Balance in Korea

PONE-D-23-04903R1

Dear Dr. Ko,

We’re pleased to inform you that your manuscript has been judged scientifically suitable for publication and will be formally accepted for publication once it meets all outstanding technical requirements.

Kind regards,

Tae-Young Pak, Ph.D.

Academic Editor

PLOS ONE

Additional Editor Comments (optional):

Reviewers' comments:

Reviewer's Responses to Questions

**Comments to the Author**

1. If the authors have adequately addressed your comments raised in a previous round of review and you feel that this manuscript is now acceptable for publication, you may indicate that here to bypass the “Comments to the Author” section, enter your conflict of interest statement in the “Confidential to Editor” section, and submit your "Accept" recommendation.

Reviewer #2: (No Response)

2. Is the manuscript technically sound, and do the data support the conclusions?

Reviewer #2: (No Response)

3. Has the statistical analysis been performed appropriately and rigorously? 

Reviewer #2: (No Response)

4. Have the authors made all data underlying the findings in their manuscript fully available?

Reviewer #2: (No Response)

5. Is the manuscript presented in an intelligible fashion and written in standard English?

Reviewer #2: (No Response)

6. Review Comments to the Author

Reviewer #2: (No Response)

7. PLOS authors have the option to publish the peer review history of their article (what does this mean?). If published, this will include your full peer review and any attached files.

Reviewer #2: **Yes: **Taehyun Ahn

---

## [Editor Report · Acceptance letter]

7 Nov 2023

PONE-D-23-04903R1 

Differential Impacts of Reduced Worktime on Work-Life Balance in Korea 

Dear Dr. Ko:

I'm pleased to inform you that your manuscript has been deemed suitable for publication in PLOS ONE. Congratulations! Your manuscript is now with our production department. 

Kind regards, 

on behalf of

Tae-Young Pak 

Academic Editor

PLOS ONE